# Synergistic Effect of Surfactant on Disperser Energy and Liquefaction Potential of Macroalgae (*Ulva intestinalis*) for Biofuel Production

Rinsha Puthiya Veettil [1,†], Rabia [1,†], Dinesh Kumar Mathew [2], Rashmi Gondi [1], Kavitha Sankarapandian [3], Meganathan Kannan [1], Gopalakrishnan Kumar [4,5], Siham Y. Al-Qaradawi [6] and Rajesh Banu Jeyakumar [1,*]

1   Department of Biotechnology, Central University of Tamil Nadu, Thiruvarur 610005, Tamil Nadu, India
2   Department of Civil Engineering, Saveetha School of Engineering, Saveetha Institute of Medical and Technical Sciences (SIMATS), Chennai 602105, Tamil Nadu, India
3   Department of Civil Engineering, Anna University Regional Campus, Tirunelveli 627007, Tamil Nadu, India
4   Institute of Chemistry, Bioscience and Environmental Engineering, Faculty of Science and Technology, University of Stavanger, PO Box 8600 FORUS, 4036 Stavanger, Norway
5   School of Civil and Environmental Engineering, Yonsei University, Seoul 03722, Republic of Korea
6   Department of Chemistry and Earth Sciences, College of Arts and Sciences, Qatar University, Doha P.O. Box 2713, Qatar
*   Correspondence: rajeshbanu@cutn.ac.in
†   These authors contributed equally to this work.

**Abstract:** The objective of this study was to evaluate the effect of surfactant on disperser homogenization pretreatment for macroalgae (*Ulva intestinalis*) to enhance biogas production. The macroalgae are subjected to surfactant coupled disperser pretreatment, which enhanced the liquefaction and improved the biomethane production. The outcome of this study revealed that 10,000 rpm at 20 min with a specific energy input of 1748.352 kJ/kg total solids (TS) are the optimum conditions for surfactant disperser pretreatment (SDP), which resulted in the liquefaction rate of 20.08% with soluble organics release of 1215 mg/L and showed a better result than disperser pretreatment (DP) with a liquefaction rate of 14%. Biomethane production through the SDP method was found to be 0.2 g chemical oxygen demand (COD)/g COD, which was higher than DP (0.11 g COD/g COD). SDP was identified to be a synergetic pretreatment method with an energy ratio and net profit of about 0.91 and 104.04 United States dollars (USD)/ton, respectively.

**Keywords:** *Ulva intestinalis*; disperser; surfactant; liquefaction; methane

## 1. Introduction

Technology advancements and rapid population growth increase energy needs. Today, fossil fuels are the most widely used energy source [1]. However, the frequent use of fossil fuels results in high levels of hazardous pollution and carbon emissions, which significantly impact climate change [2]. The community and the environment are both impacted by energy usage, whether it comes from conventional or renewable sources [3]. Therefore, it is important to move from fossil fuels to eco-friendly fuels that have been promising sustainable green energy, since they reduce pollution. More significantly, they will help to reduce the carbon footprint. The production of first-generation biofuel from food sources such as grain, maize, or soybeans led to issues with food–energy competition and high land requirements. Rich lignocellulosic feedstocks were used to produce the second generation of biofuels; however, these procedures are expensive and do not provide much of a competitive advantage over fossil fuels on the market. Compared to lignocellulosic feedstocks, marine biomass has comparatively high yields, is easily degraded with simple pretreatment, and serves as third-generation biofuels [4]. A significant source of functional metabolites, including polysaccharides, proteins, peptides, lipids, amino acids, polyphenols,

and mineral salts, may be found in marine algae [5]. Algae may naturally produce oils that may be easily blended with conventional fuel and have an energy level that is around 50% higher than ethanol. Additionally, it may reproduce within a matter of hours, making it potentially far more productive than terrestrial plants [6]. In contrast to terrestrial plants, macroalgae do not require land for cultivation but can grow quickly. This suggests that macroalgae are an effective source of biofuel [7]. *Ulva intestinalis*, a marine green alga with an unbranched thallus and a tubular frond, is generally found on rocky shores. *Ulva intestinalis* is one of the species that causes green tides, which can limit the growth of other coastal organisms and can rapidly colonize the littoral zone under favorable growth conditions [6]. *Ulva intestinalis* is also known as sea lettuce or gut weed, which has a higher concentration of carbohydrates, protein, and minerals [8].

Anaerobic digestion (AD) is a promising process for biomass treatment which converts the biomass into clean bioenergy. However, in the AD process, hydrolysis is considered to be the rate-limiting step which can be improved by implementing various pretreatment methods before the AD process [9]. Pretreatment is mainly classified as physical, mechanical, chemical, and biological treatment methods that improve the hydrolysis and AD processes. Among them, the physical and mechanical methods were considered to be efficient for the liquefaction of macroalgae [10]. Disperser pretreatment, a mechanical pretreatment, is used in this study for improving the macroalgal biomass liquefaction. By utilizing the shear force present in the probe's (rotor and stator) shear gap, the disperser aids in the successful biomass liquefaction. The internal content of the cell is released into the liquid medium as a result of cell lysis. However, the released content forms agglomeration in a liquid medium which disputes the liquefaction. This might be reduced by introducing the surfactant, which diminishes the surface tension [11]. Surfactant, the name implying the "surface active agent", reduces the interfacial tension between the liquid and solid substances, which increases the spreading of particles in a liquid medium. In this study, Triton CG-110, a non-ionic surfactant, is introduced during the disperser pretreatment. This increases the availability of organics in the liquid medium, helping in the biogas production during the AD process. By combining the surfactant with mechanical pretreatment, liquefaction may improve, with less energy spent, and enhance the AD process. Moreover, the surfactant used in this study is a biodegradable substance that is easily biodegraded when exposed to the environment. By adding a biodegradable substance to the pretreatment, this study's novel method enhances the liquefaction of macroalgal biomass and the production of biomethane.

The main objective of this study is (1) to enhance the liquefaction potential of macroalgal biomass through disperser pretreatment, (2) to optimize disperser rpm and time for effective liquefaction, (3) to study the synergistic effect of surfactant on disperser-mediated macroalgal biomass disintegration, and (4) to evaluate biomethane production using the biochemical methane potential (BMP) test.

## 2. Materials and Methods

### 2.1. Collection of Macroalgae

Ulva intestinalis, a marine macroalga, was harvested from New Mahe, Kerala, India (11°43′12.1″ N 75°31′06.1″ E). To dispose of the sand and debris, the samples were subsequently rinsed with tap water two times. The samples were then shade-dried for three days. For laboratory analysis, the materials were ground into powder and kept in a container.

### 2.2. Disperser Pretreatment (DP)

Two grams of the macroalgal sample were taken in a 500 mL beaker with 200 mL of distilled water. Disperser pretreatment was carried out using a laboratory disperser (Model: IKA T25), and the experiment was conducted at varying disperser rpm ranging from 6000 to 20,000. The effect of disperser disintegration also depends on the time, which varies from 0 to 45 min. The samples were collected at regular intervals, centrifuged, and analyzed.

### 2.3. Surfactant Disperser Pretreatment (SDP)

Surfactant disperser pretreatment (SDP) was carried out by taking 2 g of the macroalgal sample and adding to 200 mL of distilled water, which is taken in a 500 mL beaker. In the SDP process, non-ionic surfactant (Triton CG-110) was introduced into samples by varying the dosages from 0.0002 to 0.002 g/g TS at optimum disperser conditions. Samples were collected at regular intervals, centrifuged, and analyzed.

### 2.4. Biochemical Methane Potential (BMP) Test

Biochemical methane potential (BMP) is a measure of anaerobic biodegradability and is described as the portion of the compound(s) that, through the activity of microorganisms over an indefinite period of time, are transformed into biogas [12]. The BMP test was carried out in a 1L bottle with a rubber septum tightly sealing the lid. BMP experiment was carried out for control (untreated), DP, and SDP samples. In total, 200 mL of sample and 800 mL of bovine rumen fluid as inoculum were introduced in the BMP experiment bottle in a ratio of 1:3. The addition of sodium bicarbonate kept the pH between 7 and 7.4. Nitrogen gas was supplied to expel the air from the space in the bottle. The bottles were wrapped in aluminum foil to keep them in a strict anaerobic environment [13]. To ensure thorough sample mixing at room temperature, the bottles were kept in an orbital shaker at 150 rpm.

### 2.5. Analytical Methods

Total chemical oxygen demand (TCOD) and soluble chemical oxygen demand (SCOD) were determined by using the standard methods given in [14]. The presence of proteins and carbohydrates was determined by the methods outlined by [15]. Figure 1 shows the graphical representation of the study. The liquefaction rate was determined to evaluate the biomass solubilization and can be calculated as given below:

$$\text{Liquefaction (\%)} = (SCOD_1 - SCOD_2)/(TCOD - SCOD_2) \tag{1}$$

where $SCOD_1$ = SCOD release after disintegration (mg/L), $SCOD_2$ = SCOD release before disintegration (mg/L), and TCOD = TCOD concentration of the macroalgae (mg/L).

### 2.6. Energy and Cost Analysis

Energy and cost analysis was carried out to assess the effect of pretreatment in methane production to a large-scale extent. This analysis includes the input energy (applied for pretreatment, anaerobic digestion, pumping, and purification) and output energy (methane). The energy calculation and cost analysis were performed based on the work of Tamilarasan et al. [16].

#### 2.6.1. Specific Energy Calculation

To assess the cost-effectiveness of macroalgae disintegration, specific energy is a crucial metric. The following equation can be used to determine specific energy [17]:

$$\text{Specific energy (kJ/kg TS)} = (\text{Power} \times \text{Time})/(\text{Volume of sample} \times \text{Total solid}) \tag{2}$$

where Power = angular velocity × torque.

#### 2.6.2. Output Energy ($O_E$)

The methane quantity was assessed by the concentration of COD consumed, using the following equation:

$$\text{Methane production (m}^3) = \text{COD utilized (kg)} \times (0.35 \text{ m}^3/\text{kg COD}) \times \text{Biodegradability} \tag{3}$$

Output energy ($O_E$) is the energy obtained in the form of methane, and it was calculated as

$$O_E = Y \times \xi \times V \times \eta \tag{4}$$

where $O_E$ = output energy (kJ/day), Y = methane yield (m$^3$CH$_4$/m$^3$ day), $\xi$ = lower thermal value of methane (kJ/m$^3$ CH$_4$), V = working volume (m$^3$), and $\eta$ = energy conversion efficiency.

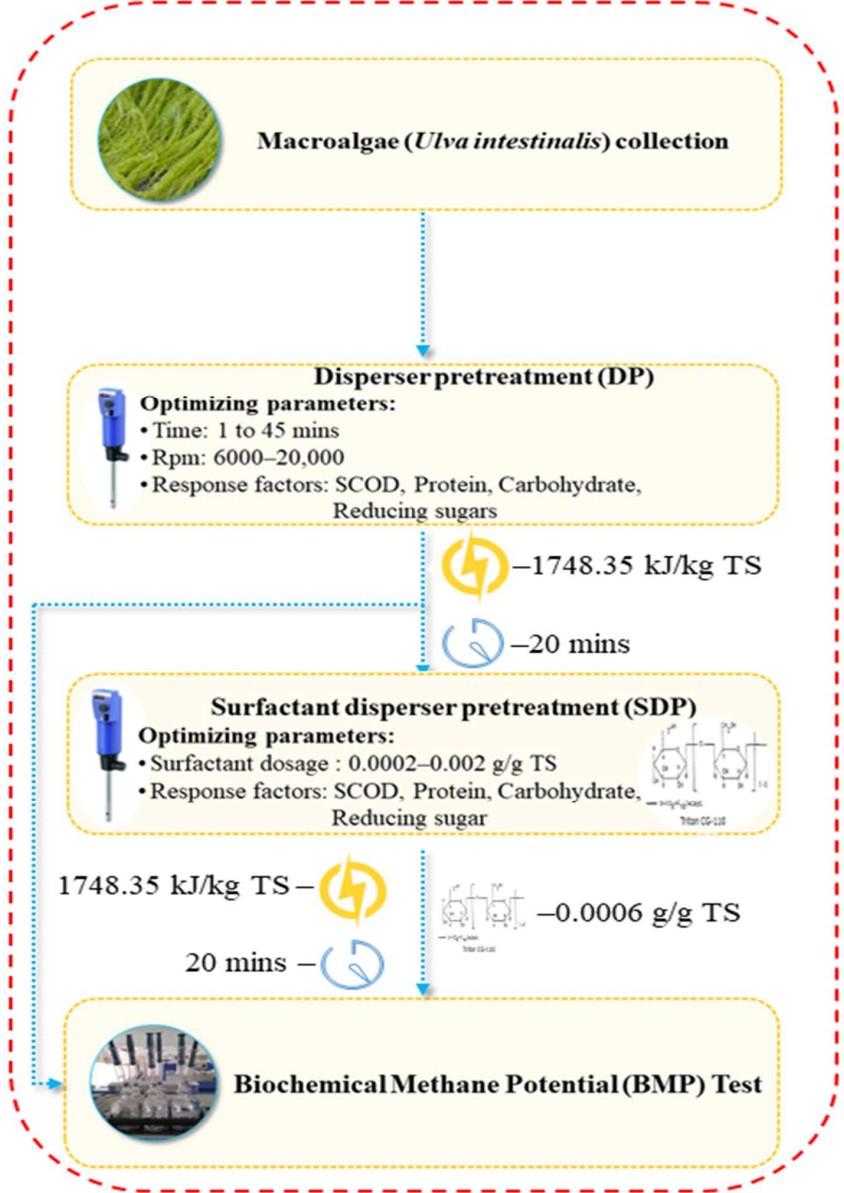

**Figure 1.** Graphical representation of the study.

2.6.3. Net Energy ($E_N$)

Net energy production was calculated by the following equation:

$$E_N = O_E - I_E \tag{5}$$

where $E_N$ = net energy (kWh), $I_E$ = input energy (kWh), and $O_E$ = output energy (kWh).

2.6.4. Energy Ratio ($E_R$)

The energy ratio was determined by dividing the output energy by the input energy as follows:

$$E_R = O_E / I_E \tag{6}$$

where $O_E$ = output energy and $I_E$ = input energy.

2.6.5. Energy Used for Pumping

Pumping energy is the energy required to pump the macroalgal biomass into the reactor from the pretreatment tank. The following equations were used to calculate the pumping energy:

$$Ep = P \times T \tag{7}$$

$$P = (\gamma \times P_C \times P_H)/\eta_P \tag{8}$$

where Ep = energy applied for pumping (kWh), T = pumping time (h), P = power needed, $\gamma$ = unit weight of the algae biomass (KN/m$^3$), $P_C$ = pump capacity (m$^3$/ S), $P_H$ = pumping height (m), and $\eta_P$ = pumping efficiency (%).

### 2.6.6. Energy Used for AD Process

The energy used for mixing in the AD process was calculated using the following equation:

$$AD_E = P_R \times V_R \tag{9}$$

where $AD_E$ = energy applied for mixing in AD, $P_R$ = power required (kW), and $V_R$ = reactor volume (m$^3$).

## 3. Result and Discussion

### 3.1. Disperser Pretreatment (DP)

### 3.1.1. Effect of Disperser Pretreatment on Organic Release

The macroalgal biomass was pretreated using a mechanical process through a disperser homogenizer. The rotor and stator mechanism underlies the operation of the disperser used for mechanical pretreatment. During dispersion, the biomass that needed to be liquefied was forced through the compressed space between the rotor and stator, pushed into the core of the probe tip, and dynamically mixed. Cavitation, viscous shear force, and turbulence are the contributory factors to the liquefaction and size reduction of macroalgal biomass. The smaller particle that liquefies was extracted through the stator teeth. Disperser pretreatment (DP) was carried out using a range of 6000–20,000 rpm and a time duration of 0–30 min. SCOD is used as a measuring index to represent the organics released during the DP and SDP processes. Figure 2 shows the release of soluble organics in different disperser rpm and times. The soluble organics released at 6000 rpm were increased from 340 to 625 mg/L by extending the liquefaction time from 0 to 20 min, respectively. The concentration of released organics remained consistent (between 625 and 800 mg/L), even when the liquefaction duration exceeded 20 min. As a result, it was hypothesized that the majority of the components that had been initially stored in the biomass would be released within 20 min, which proves that 20 min is the optimum time for mechanical pretreatment in this study. It is essential to optimize the rotor speed because the rotor speed was the key factor for optimum efficiency and effective liquefaction.

Examining the soluble organic release pattern with respect to rpm, Figure 2 reveals that the soluble organic release increases with respect to rotor speed and stabilizes [15]. Due to the partial liquefaction of the macroalgal biomass, a steady release occurs between 6000 and 8000 rpm in the disperser. This suggests that this rpm range was irrelevant to the study and is not taken into account for additional studies. A considerable increase in soluble organics release was identified when the rotor speed was increased from 9000 rpm to 10,000 rpm, implying that potential macroalgal biomass liquefaction occurs at 10,000 rpm. It was similar to the observation of [18], where the positive impact on organic release was achieved at 10,000 rpm.

Figure 3 depicts the relationship between macroalgal liquefaction and various disperser-specific energy values. From the figure, it was identified that a minimum liquefaction of 5.6% was achieved at 6000 rpm. At the optimum disperser condition, 8.3% liquefaction was identified during specific energy of 437.08 kJ/kg TS. A stable increase in liquefaction was identified in specific energy of 437.08 to 3933.79 kJ/kg TS, while the optimum liquefaction of 14.9% was achieved in 1748.35 kJ/kg TS.

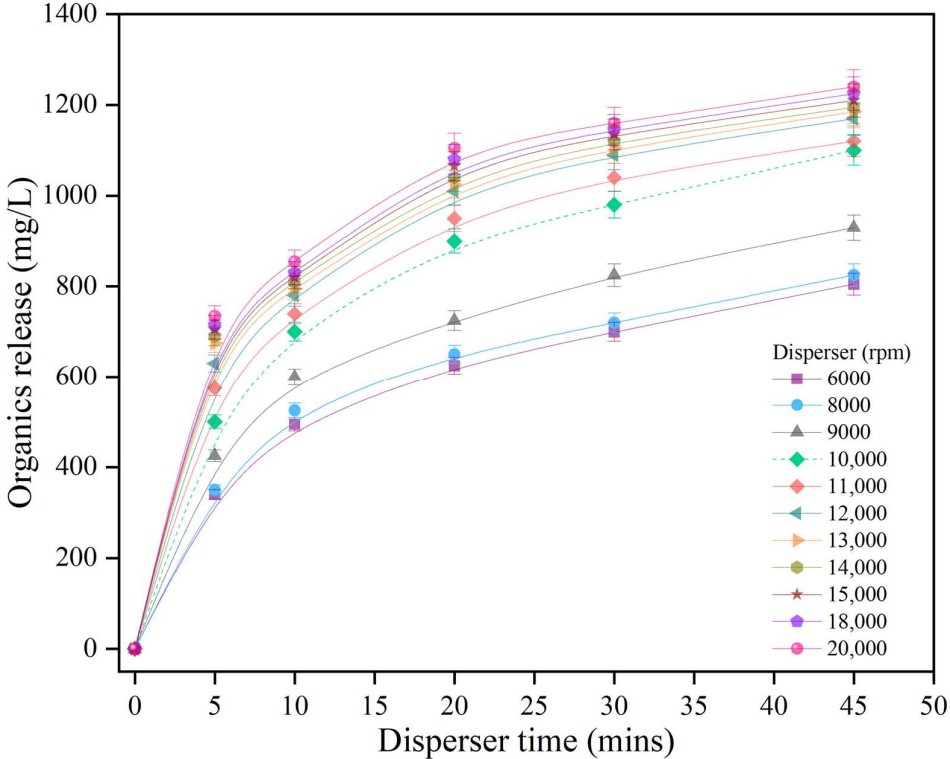

**Figure 2.** Effect of disperser time on organics release.

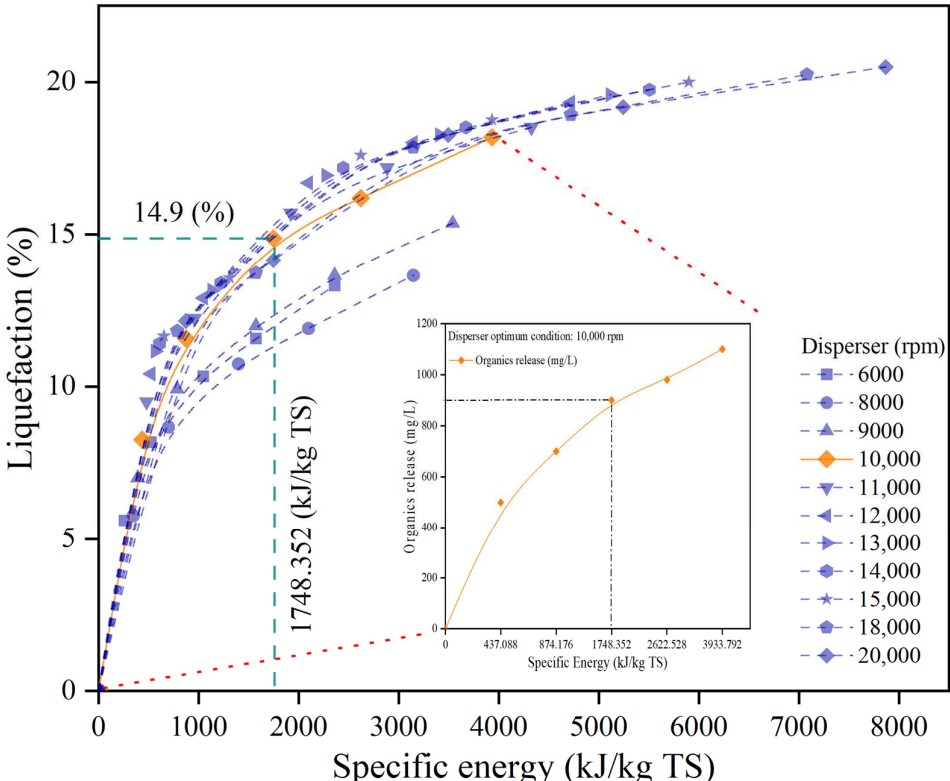

**Figure 3.** Effect of specific energy on organics liquefaction and effect of specific energy on organic release at optimal disperser condition (10,000 rpm).

From the above result, it was identified that a liquefaction time of 20 min and a rotor speed of 10,000 rpm is optimum with specific energy spent of about 1748.35 kJ/kg TS for the mechanical pretreatment process.

### 3.1.2. Effect of disperser pretreatment on biopolymer release

Proteins and carbohydrates are biopolymers that are prevalent in macroalgae biomass and may be useful in the production of biomethane. Due to the cell being intact, the availability of biopolymers is lower in biomethane production during the AD process. The availability of biopolymers might be increased through pretreatment. The effect of protein and carbohydrate release on specific energy at the disperser optimum condition is shown in Figure 4. At 437.08 kJ/kg TS, the release of carbohydrates and protein was identified as 210.5 and 85.72 mg/L, respectively. This might increase rapidly up to 480.95 mg/L (carbohydrates), 220.4 mg/L (protein), and 103.71 mg/L (reducing sugars) mg/L at specific energy spent of 1748.352 kJ/kg TS. Thereafter, no rapid increase in biopolymer releases up to 3933.79 kJ/kg TS was identified.

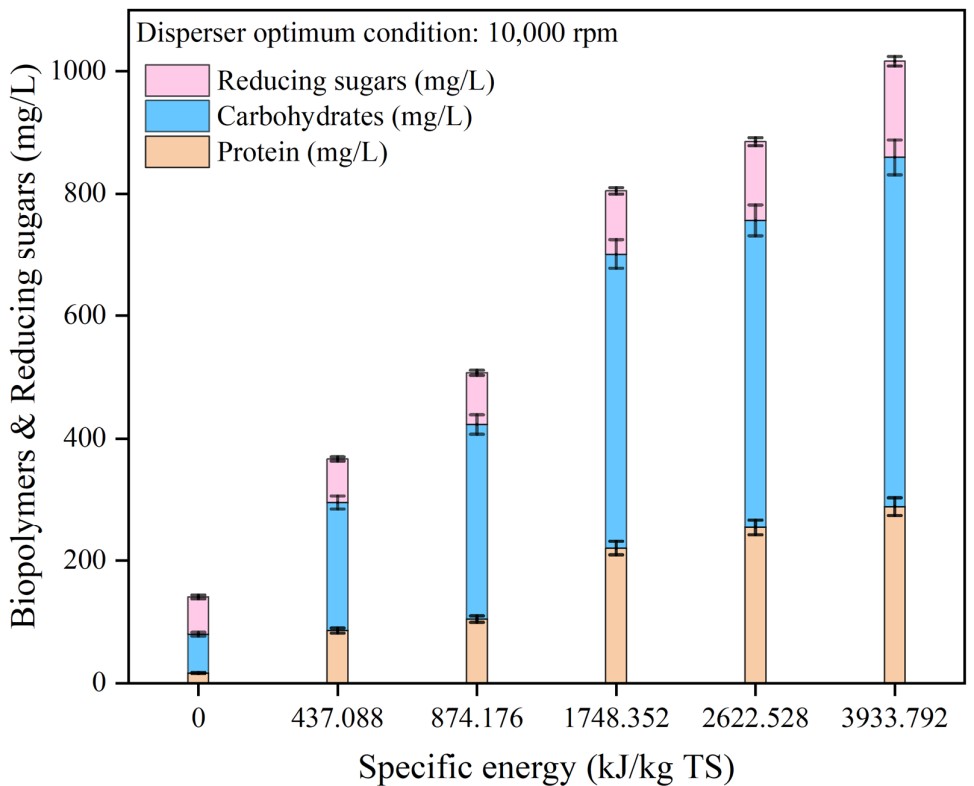

**Figure 4.** Effect of specific energy on release of protein, carbohydrates, and reducing sugars.

At the optimum disperser rpm, there was a significant release of biopolymer at 20 min. Consequently, the algal cell wall is effectively disintegrated, which improves the distribution of intracellular organic material to the aqueous phase of the media.

### *3.2. Surfactant-Induced Disperser Pretreatment (SDP)*

### 3.2.1. Effect of SDP on Organic Release

At the optimum disperser conditions (10,000 rpm) and liquefaction time (20 min), surfactant was used on the sample at varying concentrations (0.0002 to 0.0023 mg/L). Due to the macroalgal aggregation caused by the increased surface energy, disperser pretreatment efficiency is decreased. Combining surfactant with DP can be compensated. To prevent aggregation of the sample, the surfactant molecules surround the disintegrated macroalgal biomass in an organic layer and lower the surface tension [19]. The effect of surfactant on organic release is shown in Figure 5. The macroalgal biomass was treated with varying

surfactant concentrations ranging from 0.0002 to 0.0023 mg/L. Organic release occurs in two phases: a rapid phase and a stabilizing phase. The organic release increased from 900 to 1215 mg/L during the rapid phase (0.0002 to 0.0006 mg/L). The release of intracellular organics into the aqueous phase is caused by the combined impact of surfactant on disperser pretreatment, as the surfactant prevents agglomeration, which increases more liquefaction of macroalgal biomass [17].

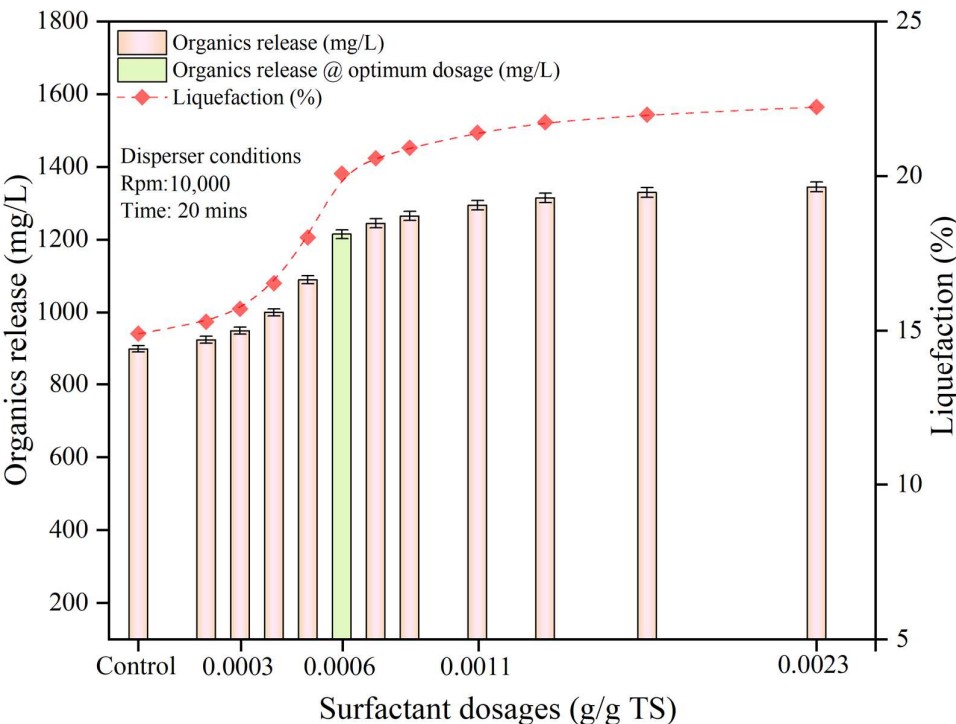

**Figure 5.** Effect of surfactant dosages on organics release in disperser optimum condition.

The organic release during the stabilizing phase (0.0008 to 0.0023 mg/L) was determined to be in the range of 1245 to 1345 mg/L, respectively. When compared to the rapid phase, a mild increase in the organic release is noticed during the stabilizing phase. For DP alone, 1748.35 kJ/kg TS of specific energy was needed to achieve a liquefaction of 14.9%, whereas, for SPD, the same specific energy can provide a liquefaction of 20%. This indicates that the surfactant concentration of 0.0006 mg/L is sufficient for the effective liquefaction of the macroalgal biomass. Surfactant concentrations above 0.0006 mg/L have no effect on the liquefaction of macroalgae but rather increase the cost of the chemical.

### 3.2.2. Effect of SDP on Biopolymer Release

According to Mitra et al. [20], the *Ulva intestinalis* species contains a significant amount of protein and carbohydrates, which is useful in biogas production. The greater carbohydrate fraction contains easily soluble polysaccharides that can be hydrolyzed into monosaccharides such as glucose or mannose. Proteins can also be similarly hydrolyzed into basic amino acids. These simple substances are readily fermentable and can be utilized by methanogens [21]. The effectiveness of methane synthesis is increased by the presence of these biopolymers. The surfactant promotes liquefaction by lowering surface tension and preventing particle agglomeration [19]. The release of the biopolymers occurs in two stages: a rapid phase and a stabilizing phase. Figure 6 shows that the rapid phase is marked by a considerable release of biopolymer, as the surfactant concentration increased from 0.0002 to 0.0006 g/g TS. The trend for protein, carbohydrates, and reducing sugars is the same during rapid release with the concentration of 165.5 to 374.5 mg/L (protein), 413 to 645 mg/L (carbohydrates), and 103.71 to 217.35 mg/L (reducing sugars). This rapid

increase in biopolymer release is due to the synergetic impact of surfactant in disperser treatment. The biopolymer release becomes stable with no such noticeable increase during the stabilizing phase, which lasts from 0.0008 to 0.002 g/g TS with the release of 378.4 to 392 mg/L (protein), 650.95 to 664.4 mg/L (carbohydrates), and 217.35 to 228.05 mg/L (reducing sugars).

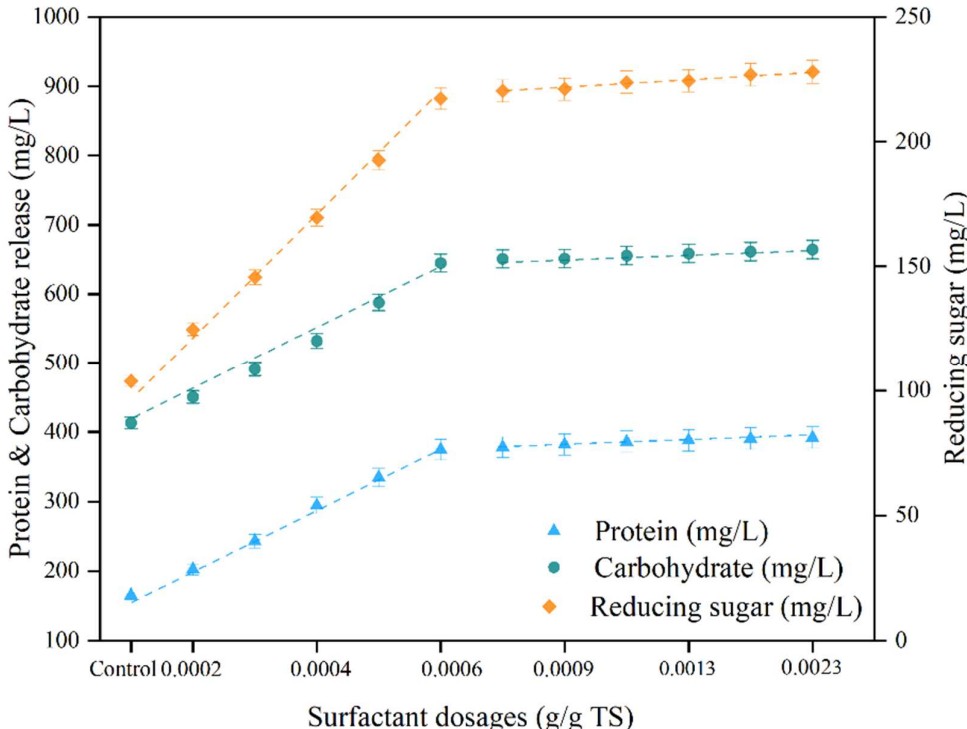

**Figure 6.** Effect of surfactant dosage on the release of protein, carbohydrates, and reducing sugars.

Thus, it was revealed that SDP was very effective at liquefying macroalgal biomass and releasing biopolymers.

### 3.3. Biomethane Production

The biomethane potential experiment was carried out for control (untreated) and pretreated (DP and SDP) samples to estimate the methane production. The AD process becomes extended and limited due to the rigid structure of the biomass. An effective disintegration process can break down the cell's structure and release internal molecules that boost methane production.

In this study, DP and SDP are performed to disintegrate the macroalgae biomass. Methane production for all the samples is shown in Figure 7. From Figure 7, it was identified that at the beginning and end of the biodegradation process, methane generation was very low and that was correlated with methanogenic bacterial growth. At the initial stage, methane production was noticed as 0.005, 0.035, and 0.095 g COD/g COD. Methane production was low in the initial days, and this is because the microbes in the inoculum are not well accommodated yet and need more time to degrade the substrate. After being well acclimatized, methane production started to upsurge significantly with the increase in the biodegradation period up to 15 days, attaining higher methane production of 0.2 g COD/g COD in SDP compared to 0.03 (control) and 0.11 g COD/g COD (DP), respectively. This might be because anaerobic microbes are hydrolyzing soluble organic materials more frequently, which promotes increased biomethane production. In contrast, the control sample recorded a very insignificant amount of methane (0.03 g COD/g COD), this is due to the complex structure of macro polymers, which takes longer for the methanogens to degrade. In SDP samples, the maximum methane production is obtained due to the

availability of biopolymers in the digestion period compared to disperser-treated and untreated samples. From the above discussion, it was concluded that combined pretreatment improves methane production compared to control and disperser pretreatment.

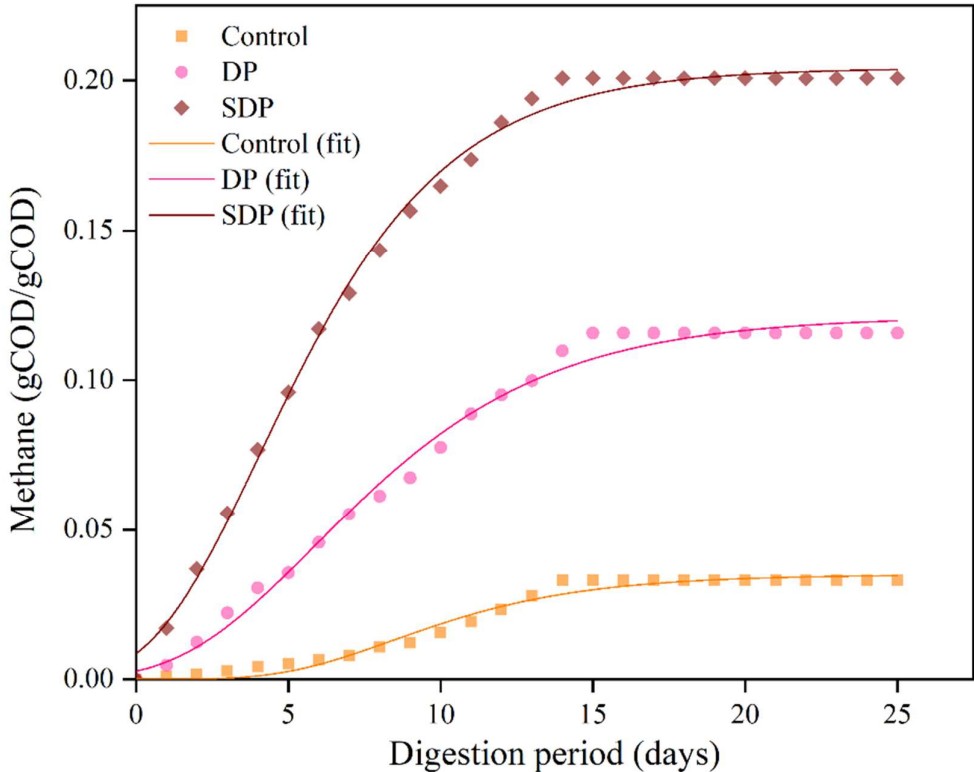

**Figure 7.** Methane production in various samples.

Methane production was found to be stabilized after 25 days, due to substrate diminution. Among the samples, methane production efficiency was performed by using first-order kinetics fit. The observed $R^2$ was found to be 0.97–0.99, which indicates the actual data come under the best fit.

*3.4. Energy and Economic Assessment of DP and SDP*

An absolute economic analysis must be performed based on the experimental data and cost information to evaluate the factual suitability of employing DP and SDP. One ton of macroalgae biomass was considered as an input parameter in this study. For estimating the cost–benefit analysis of a pretreatment process, economic efficiency and energy calculations were key considerations. To compare the economic viability of implementing the process, operational cost analyses for both DP and SDP were conducted. COD solubilization is used as an index for economic analysis of pretreatment methods. The energy and cost analysis related to the pretreatment and methane production were analyzed based on the equations that were discussed in Section 2.6, and the analysis is shown in Figure 8. In this study, the energy ratio was calculated as 0.37 and 0.91 for DP and SDP, respectively, which showed that applying SDP on large scale might be feasible. In terms of net profit, SDP achieved a higher net profit of 104.04 USD/ton of marine macroalgae compared to DP (39.15 USD/ton of marine macroalgae), respectively. In this study, energy and cost demand were less in SDP compared to DP pretreatment. Based on the above discussion, the SDP process is considered to be economically feasible to implement on a large scale.

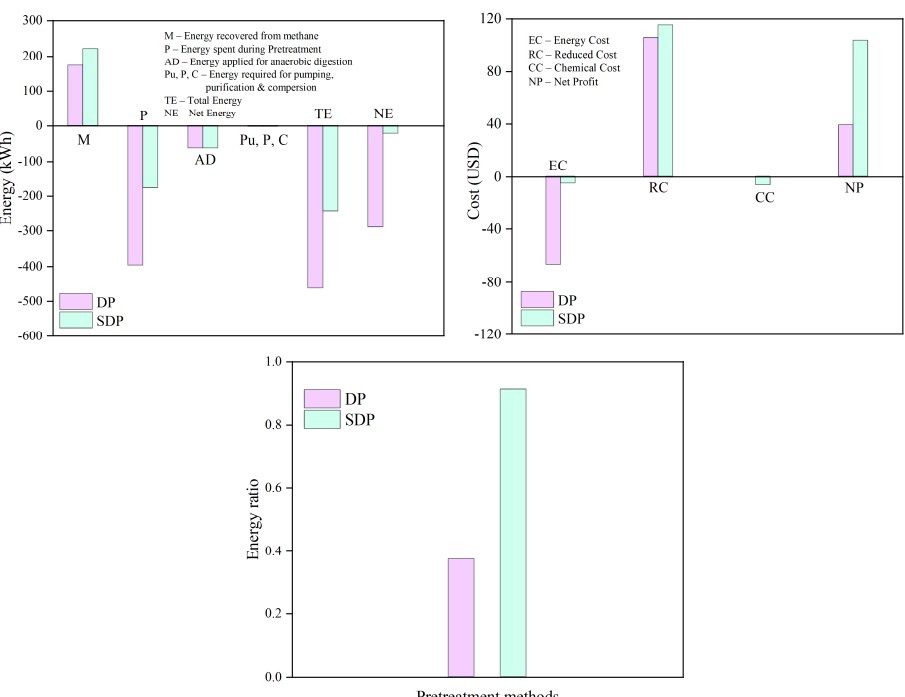

**Figure 8.** Energy and cost analysis for DP and SDP processes.

## 4. Conclusions

The present study revealed the effect of surfactant on a disperser-pretreated marine macroalgal biomass that can be utilized as a potential feedstock for biomethane production. A COD liquefaction of 14.9 % was achieved at a disperser rpm of 10,000 in 20 min of liquefaction time with a specific energy input of 1748.352 kJ/kg TS. Then, the surfactant was introduced into the DP process at the dosage of 0.0006 g/g TS, which improved the liquefaction by about 20%. At the optimal conditions, SDP samples showed a higher methane yield of 0.2 g COD/g COD compared to DP (0.11 g COD/g COD). Energy analysis revealed that SDP showed an energy ratio of 0.9 greater than DP (0.3). The above results revealed that the synergistic effect of surfactant on disperser pretreatment was concluded as a promising liquefaction technique for macroalgal biomass and methane production.

**Author Contributions:** R.P.V. and R.: wet chemical analysis and writing; D.K.M. and R.G.: analysis of data and writing; K.S.: conceptualization; M.K.: resources; G.K.: resources; S.Y.A.-Q.: Resources; R.B.J.: supervision, data validation, and project lead. All authors have read and agreed to the published version of the manuscript.

**Funding:** This work is supported by Department of Biotechnology, India under its initiative Mission innovation Challenge Scheme (IC4). The grant from the project entitled "A novel integrated biorefinery for conversion of lignocellulosic agro waste into value added products and bioenergy (BT/PR31054/PBD/26/763/2019) is utilized for this study.

**Institutional Review Board Statement:** Not applicable.

**Informed Consent Statement:** Not applicable.

**Data Availability Statement:** Not applicable.

**Conflicts of Interest:** The authors declare no conflict of interest.

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
