# Peer review of "Synergistic Effect of Surfactant on Disperser Energy and Liquefaction Potential of Macroalgae (Ulva intestinalis) for Biofuel Production"

_fermentation, doi:10.3390/fermentation9010055_

Round 1

Reviewer 1 Report

The work is very interesting and well-written but chapter 3.4. needs to be completed with methodology. It is unclear how the Authors counted energy inputs and costs on a large scale. Please keep in mind the effect of scale. How were the energy calculations done on a large scale? Please complete and add this aspect to the summary.

Reviewer 2 Report

The utilization of marine macroalga as starting material to produce energy material is a hot topic under the present situation of resource shortage and environment pollution. In this manuscript, the authors comparatively studied the pretreatment methods on the production of methane form Ulva intestinalis, a marine macroalga. It was found that the combination of disperser homogenization pretreatment and the use of non-ionic surfactant (Triton CG-110) could enhance the release of organics and then the production of methane. LCA analysis was also performed. Before acceptance of it for publication, the followings should be addressed.

1.       The use of abbreviation should be firstly defined, for example, “TS” appeared in the abstract has not been defined, page 2, line 62, “AD”.

2.       Page 3, lines 125-129, this part should be carefully checked. First of all, the explanation of the signs does not match with the formula, which is also needed to be checked. Secondly, the authors used the term “the concentration of the biomass”, which needs to be explained a little bite more. Is it the concentration of macroalga ? How to get its value?

3.       The quantification of the organics release should be given. If the COD is used to represent the organics release, it should be stated explicitly to make the manuscript be harmonious.

4.       The authors are suggested to consider the reasons for the differences in the methane production from the three different sources.(Figure 7). Are the differences caused merely by the different concentrations of feedstock? Or some other reasons exist?

5.       Proof reading should be carefully checked. Typical examples are:

(1)    Page 2, line 55-56, please complete the statement, “Ulva intestinalis, a marine green alga with unbranched thallus and a tubular frond.”, to make it be a sentence or be combined to another sentence.

(2)    Page 2, lines 64-65, please check if the sentence, “Pretreatment are classified as physical, mechanical, chemical, and biological treatment methods.”, needs to be improved.

(3)    Page 2, lines 77-78, the statement, “By combining the surfactant with mechanical pre-treatment lessen the energy requirement and improves the liquefaction.”, needs to be improved.

(4)    Page 6, lines 188-189, the sentence, “Proteins and carbohydrates are biopolymers which present abundantly in macroalgal biomass, may useful in biomethane production.”, needs to be improved.

(5)    Page 6, line 194, the phrase, “This might be increases rapidly up to 480.95 (carbohydrate),”, needs to be refined.

(6)    Page 9, lines 258-260, the sentence, “From the figure, it was identified that at the beginning and end of the biodegradation process, methane generation was very low that are correlated with the methanogenic bacterial growth.”, needs to be improved.

Round 2

Reviewer 2 Report

The authors have addressed my concerns carefully, and now the manuscript is improved. I recommend acceptance of it for publication in this journal.